# What facilitates or prevents academic fraud in a Colombian faculty of medicine–Protocol of a study using fuzzy cognitive mapping

**Juan Pimentel**[1], **Paola López**[1], **Johan Rincón**[1], **Laura Neira**[1], **Daniel Jiménez**[1], **Camilo Correal**[1¤]*, **Iván Sarmiento**[2,3]

**1** Department of Family Medicine and Public Health, Universidad de La Sabana, Chía, Colombia, **2** CIET-PRAM, Department of Family Medicine, McGill University, Montreal, Canada, **3** Escuela de Medicina y Ciencias de la Salud, Universidad del Rosario, Bogotá, Colombia

¤ Current address: Department of Family Medicine and Public Health, Universidad de La Sabana, Chía, Cundinamarca, Colombia
* camilo.correal@unisabana.edu.co

**Data Availability Statement:** No datasets were generated or analysed during the current study. All

## Abstract

### Introduction

Academic fraud is any behavior that gives a student an undeserved advantage over another student. Few studies have explored the causes of and possible solutions to academic fraud in Latin America. We aim to map the knowledge of stakeholders in a Colombian faculty of medicine about the factors that facilitate and prevent academic fraud.

### Methods

Fuzzy cognitive mapping. We will use the approach proposed by Andersson and Silver to generate fuzzy cognitive maps representing stakeholder knowledge. This process consists of ten steps: (1) definition of the research question; (2) identification of participants; (3) generation of ideas; (4) rationalization of ideas; (5) organization and connection of ideas; (6) weighing; (7) pattern grouping; (8) list of links and digitization; (9) combination of maps and network analysis; and (10) deliberative dialogue. To draw the maps, we will invite medical students, interns, resident physicians, master's students, and professors in the faculty of medicine. Four medical students will receive training to facilitate the sessions. Participants will identify the factors contributing to academic fraud and their causal relationships. We will use a combination of network analysis and graph theory to identify the chains of factors with greatest influence on academic fraud.

### Conclusion

The maps will serve to discuss strategies to reduce academic fraud in the Faculty of Medicine and to identify factors that could be addressed in other contexts with similar problems. This research will allow the students who facilitate mapping sessions to learn about research techniques, fuzzy cognitive mapping and academic fraud.

relevant data from this study will be made available upon study completion.

**Funding:** This study is supported by La Sabana University (grant number: 860075558-1). The funders had and will not have a role in study design, data collection and analysis, decision to publish, or preparation of the manuscript.

**Competing interests:** The authors have declared that no competing interests exist.

**Study registration:** Registered in OSF Registries on August 2nd, 2022. Registration number: osf.io/v4amz

## Introduction

Although there is no globally accepted understanding of academic fraud [1], a widely used definition is any behavior that gives a student an unearned or undeserved advantage over another student [2, 3]. Among the health sciences, academic fraud includes copying others, using false clinical notes, sharing information about exams, and lying about physical exams performed on patients [4]. Fraud can be active when the goal is to gain a benefit for the student themselves or passive when the goal is to help a partner [5]. Since there are important differences between medical schools around the world, the definition of fraud must be flexible, progressive, and tailored to the local context in which they operate [6, 7]. For example, fraud practices have evolved as technology has come to play a critical role in academic life [7], which has been particularly important in the context of the Covid-19 pandemic.

Academic fraud is a frequent phenomenon. For example, the prevalence of fraud has been estimated to be as high as 58%in medical schools in the United States [4]. Unfortunately, academic fraud in the health sciences leads to potentially serious consequences. In medicine, academic fraud during student life has been reported to be associated with disciplinary problems during professional life [8]. Fraud also has economic consequences. In 2012, the U.S. Institute of Medicine estimated that the National Health System loses $765 billion annually due to administrative malfunction, which includes $75 billion due to fraud [6]. An example of fraud is the case of California doctors issuing bills for the same drugs multiple times [9].

In Colombia, a study based on data collected from 2003 to 2013 among four top universities found that more than 90% of students admitted having engaged in fraud in their academic life [7]. This study also identified types of academic fraud committed in the country, such as letting another student copy on an exam, including the name of a student who did not work in the assignments, lending an assignment (passive fraud), impersonation in an exam, downloading assignments from the internet, or using specific tools (like calculators) when it was not allowed (active fraud).

The factors promoting academic fraud have been also classified as intrinsic and extrinsic [10]. Intrinsic factors are characteristics such as gender, self-esteem, level of maturity, moral development, and academic performance. The extrinsic factors comprise the systems of institutional (legal) control, as well as the cultural and moral environment. Martínez et al. [7] described the perception of cost-benefit facilitating fraud: students perceive a low risk of being detected or sanctioned and simultaneously important benefits if they succeed in their attempt. In Colombia [7], these researchers reported reasons why students commit fraud such as concerns about the grade point average, memory-oriented exams and evaluations, poor quality of teaching, a poorly understood exam, burnout, and a desire to help a friend. Additionally, factors such as the desire to learn, considering fraud as something dishonest, and feeling bad about committing fraud were identified as protectors against fraud.

Different authors and monitoring entities agree that health professionals, including doctors and medical students, should receive training in academic integrity to prevent fraud [6, 11]. In the United States, for example, the objective of these programs is to prevent fraud, abuse, errors and waste of resources [6]. Similarly, there is consensus that training programs should be directed simultaneously at medical students and professors, to achieve a synergistic effect [11].

Little research has addressed the causes and consequences of academic fraud in Latin America [7]. The few existing fraud prevention programs, much more common in North America, are focused on educating healthcare professionals on administrative aspects related to paying reimbursements for services [6, 11]. This study aims to map the knowledge of stakeholders in a Colombian faculty of medicine about the factors that facilitate and prevent academic fraud.

## Methods

### Setting

We will conduct the study at the Faculty of Medicine of La Sabana University in the municipality of Chía, Colombia. Chía is located 15 km from Bogotá, the capital of Colombia. La Sabana University is a private institution that currently offers 23 undergraduate and 99 graduate programs [12].

The university has 8,926 undergraduate students. Some 22% of these students come from a low socioeconomic background, 52% belong to the middle class, and the remaining 26% come from a high socioeconomic level [12]. Currently, there are about 900 students enrolled in the Faculty of Medicine, where the undergraduate medical program lasts seven years.

### Study design

Study uses fuzzy cognitive mapping (FCM), which are commonly used to explore the constellation of factors that contribute to outcomes and to inform health decisions. FCM comprises concepts or nodes (determinants or factors that influence a system of additional factors) and causal links (connections between nodes) which receive a weight according to their relative strength (Fig 1) [13].

### Study procedures

We will follow the steps proposed by Andersson and Silver [13] to generate fuzzy cognitive maps representing the factors that facilitate or prevent academic fraud. This process consists of ten steps: (1) definition of the research question; (2) identification of participants; (3) generation of ideas; (4) rationalization of ideas; (5) organization and connection of ideas; (6) weighing; (7) pattern grouping; (8) list of links and digitization; (9) combination of maps and network analysis; (10) deliberative dialogue. We describe each of the steps below.

**Research question.** What factors promote or prevent academic fraud at the Faculty of Medicine of La Sabana University?

We will use the definition of academic fraud proposed by Genereux et al. [5] and taken up by Eshet: [3] any behavior that gives a student an unearned or undeserved advantage over another student. The definition is broad and doesn't distinguish between severity levels of infractions. Infractions' seriousness can vary based on circumstances and subjective interpretations. Nonetheless, the definition allows participants to align their views with what they consider fraudulent behavior.

**Participants.** Andersson and Silver [13] recommend organizing participants into homogeneous groups of no more than five people to avoid power imbalances and to facilitate discussion in the mapping sessions. We will invite five groups, each consisting of five people, resulting in a total of at least 25 participants. We will make successive maps with additional groups of stakeholders until we achieve qualitative saturation of concepts. Participants will be organized and separated by groups of stakeholders as follows: students from preclinical semesters (I to IV); students from clinical semesters and medical interns (V to XIV); resident

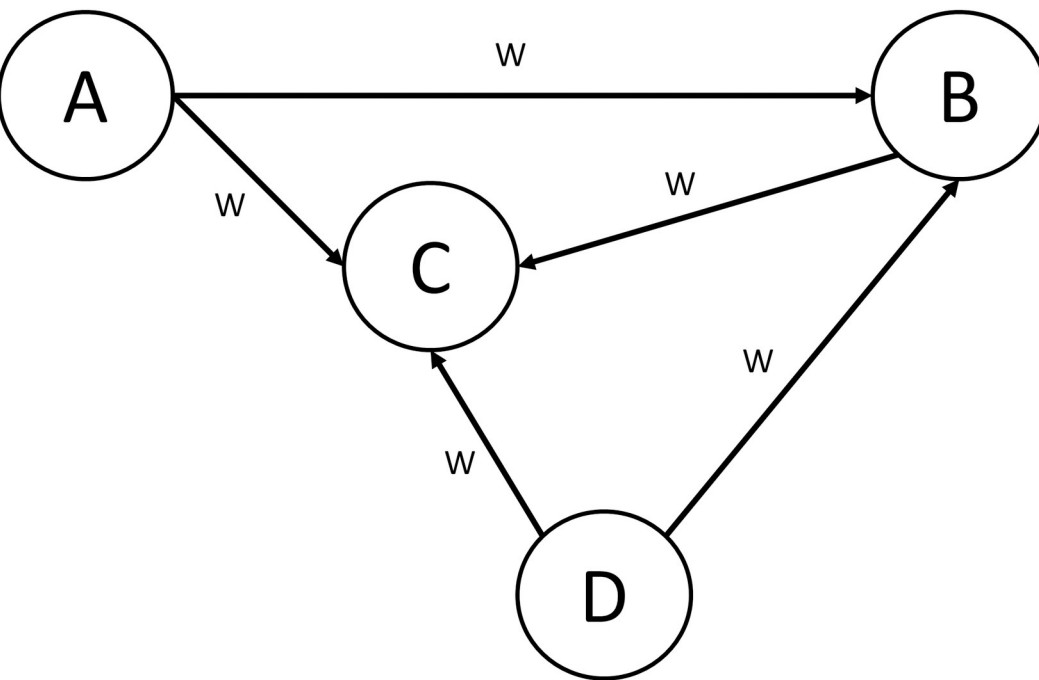

**Fig 1. Representation of a fuzzy cognitive map.** The map comprises concepts or nodes (A, B, C, and D) and weighted causal links (W).

physicians; master's students; and professors of the faculty of medicine.The inclusion criteria are 1. Being a student, intern, resident, master's student, or professor of medicine of legal age (≥18 years) at the faculty of medicine of La Sabana University; and 2. Having received at least two doses of any COVID-19 vaccine approved in Colombia.

The Faculty of Medicine will contribute to participant recruitment by issuing a call through their message list (S1 File). All students and faculty members associated with the faculty will receive the invitation. Those who wish to take part in the study will register on an online page accessible only to a member of the research team (JP). During registration, each participant will propose their preferred schedule, and the groups will be organized accordingly.

The lead author (JP) will train four research assistants as facilitators of the FCM sessions using the resources found on https://www.ciet.org/fcm. These include an instructional video located at https://www.screencast.com/t/6whTNuRun6W that provides detailed information on how to conduct the sessions. The trainees will conduct supervised practice sessions with volunteers to ensure they adhere to the standardized protocol. The analysis will not include the practice maps. Once the trainees do not need additional corrections, they will start data collection.

Each session will have a facilitator and a reporter to take notes of the discussions. For each session, the number of participants and their characteristics in terms of gender, age, place of origin, semester, and academic background will be recorded.

**Mapping sessions.** Each mapping session will follow these steps:

*1. Generation of ideas*

Using a sheet of whiteboard and *post-its*, the facilitator will place a central node "academic fraud" in the center of the whiteboard and will ask the participants: what does academic fraud mean to you?

After all the participants answer the question, the facilitator will ask what do you consider to be the factors that promote or prevent academic fraud? Participants will have 15 minutes to write concepts in the *post-it*s and will add the concepts that positively or negatively influence the occurrence of the central node. This step may take less time if participants exhaust the concepts sooner than expected.

*2. Grouping of ideas*

The facilitator and participants group related concepts and eliminate duplications. For example, words that mean the same thing but are expressed differently (cell phone and smartphone). This step will help synthesize and clarify the map. Participants will have to reach a consensus and accept each new category created. This step will also be an opportunity to eliminate, redefine, deepen, or even suggest new concepts.

*a. Connection and organization of nodes*

In this step, participants will draw the connections between concepts using arrows to indicate the direction of the cause-effect relationship. Connections can be between concepts and the central node, or between concepts. This step will also be an opportunity to redefine concepts, reshape and improve the map.

*b. Weighing*

Participants will assign a weight to each relationship between nodes to indicate variability in levels of influence. This scale uses numbers to indicate the strength of the relationship and positive or negative signs to represent the direction of its influence. Positive values indicate that the increase of a node will increase another node and negative values indicate the opposite. Weights will be expressed in the range of 1 to 5. Values closer to zero correspond to a weaker level of influence and values closer to five indicate stronger levels of influence.

To establish the weights, once all the arrows are drawn the facilitators will ask: which of all these relationships is the strongest? In other words, on a scale of 1 to 5, what relationship would be a 5? Then the facilitators will ask: which of all these relationships is the weakest? In other words, on a scale of 1 to 5, which relationship would be a 1? After this, the facilitators will ask the participants about the relative weight of the remaining arrows. If there are disputes or differences of opinion between participants, the reasons will be noted and the consequence of each of the alternatives will be established in the final model.

Once all the arrows have received a weight validated by all participants, a photograph of the map will be taken. On the whiteboard, the name of the group, its members, and the date will be noted.

**FCM analysis.**

*1. Digitization and list of relationship*

An edge list will be created in the form of a table with three columns: cause, effect, and weight. Subsequently, we will calculate fuzzy transitive closure to establish the strength of the influence that one node has on others through direct or indirect relationships [14]. We will use the free software CIETmap to apply this model [15]. The fuzzy transitive closure maps will be transferred to the freeware yEd [16] for visualization and further analysis.

*2. Comparison of patterns*

A pattern matching table will be constructed to identify which concepts are similar or dissimilar across the maps.

*3. Map combination and network analysis*

We will combine maps by stakeholder group. The weights of the relationships in the group map will be the average of their weights in the individual maps. The final report will also include a measure of the dispersion around the mean as an indicator of agreement within the group. yEd provides measures of network analysis, such as the weighted out-degree centrality of a node, which identifies nodes with important role as causes on the map. In-degree centrality uses the incoming relationships to identify important outcomes on the map.

**Deliberative dialogue.** We will organize meetings with each group of participants to communicate the results and to discuss recommendations regarding potential interventions to reduce the incidence of academic fraud. These meetings will be also an opportunity to conduct member-checking with stakeholders. At verification meetings, research results will be shared with participants to ensure that the information reported matches their experiences and opinions [17]. Participants will then discuss maps and tables to identify priority areas to be addressed. During these sessions we will discuss potential differences in the interpretation of findings according to the severity of fraud. This process, which will be an intervention itself, will generate recommendations for action. The results from deliberative dialogue will be useful to propose interventions that we can explore with more robust designs such as a randomised controlled trial.

## Ethical considerations

We are aware that research with students may raise concerns regarding coercion, confidentiality of data, and anonymity of participants. However, we also recognize that participating in a study has an educative value for students, by exposing them to innovative research methods used in their discipline and involving them in the collection and analysis of their own data [18].

According to the University of Alberta's Guidelines for Conducting In-Class Research [19], research including students is considered ethical, provided that: a) The methodology and purpose of the study are approved by the teacher whose students will be participating; b) The study is approved by the head of the unit for its educational value to the participating students; c) Each student has permission to participate or decline participation anonymously; d) The methodology preserves the confidentiality of individual student responses; e) Students are invited to a debriefing meeting after the completion of the study; f) The research complies with the Tri-Council Policy Statement of Canada [20].

This study will apply all the aforementioned recommendations and will follow the ethical principles of the Canadian Institutes of Health Research Tri-Council Policy Statement [20]. Before starting each session, we will obtain the informed consent of the participants. The risks and benefits may differ between different groups of study participants, so consent forms will have information on risks and benefits specific to each group of participants (medical students, residents, master's students, and faculty). Research assistants will explain the purpose of the study, the confidentiality of the answers, and the rights of respondents not to answer certain questions or to terminate their participation in the study. The project was approved by the Research Ethics Committee of La Sabana University on 28th, March 2022.

## Safety considerations

We recognize that the topic covered in this research (academic fraud) is sensitive to participants. However, the research explores the facilitating and preventing factors of fraud, and will not inquire about personal experiences of fraud that participants may have incurred in. We

will kindly remind participants not to share any personal experiences of fraud. To guarantee confidentiality, we will not record the sessions, and the note taker will not include any information that could allow us to identify the identity of individuals.

This study employs non-invasive techniques. Based on the above, according to resolution No. 8430 of 1993 of the Ministry of Health of Colombia, the study poses minimal risk to the participants [21]. Similarly, the research is classified as of minimum risk according to the Tri-Council Policy Statement of Canada [20]. This means that the likelihood and magnitude of potential harm from their participation in a study are no greater than those found by the participant in those aspects of their daily lives that relate to the research.

To avoid possible undue influence or coercion with students, a third party unrelated to the research, who has no power or authority over the students, will be present for the consent process. The consent will explain that no sanctions (academic or otherwise) will be generated for not agreeing to participate in the study.

Finally, we will make sure to implement strict measures to prevent the spread of COVID-19. We will only invite participants who have received at least two doses of any COVID-19 vaccine approved in Colombia. Mapping sessions will be held observing hand hygiene practices every 15 minutes. We will ask participants to always wear masks and to maintain a minimum distance of two meters. Finally, each student will use their own materials (pens, post-it) and the circulation of materials among the participants will be discouraged.

## Data management plan

When digitizing information, we will treat all participant responses as confidential, and we will not record names or identify information along with their responses. A computer with a password will protect the encrypted data, and the physical copies will be secured in a locked cabinet in the lead author's office at La Sabana University, inaccessible to anyone outside the study. The original paper records will be transported, stored, retained and, after seven years, destroyed following the guidelines of the Center for Research in Tropical Diseases (CIET) [22] for the security, storage and eventual destruction of paper records.

## Dissemination plan

This protocol won the "First Challenge—Faculty of Medicine", an academic competition held at the Faculty of Medicine of the La Sabana University [23]. The *Challenge* is an innovation contest aimed at promoting collaboration between professors, students and staff to address academic problems. In its first version, the contest focused on the topic of academic integrity, proposing the following challenge: How could academic integrity be promoted to prevent fraud at the La Sabana University?

We will organize semestral meetings with the academic community to communicate the status of our project. This project will be presented in at least two international conferences. For example, medical student PL presented an oral presentation in the 2022 The Network: Towards Unity for Health conference in Vancouver, Canada [24]. We will publish at least two articles in peer-reviewed journals.

## Status and timeline

We started the study recruitment in August 2022. Three medical students and nine master's students in public health have participated in three mapping sessions (one with the medical students and two with the master's students). We expect to complete the anticipated sample size by the beginning of 2023.

## Discussion

This study will shed light on the factors associated with academic fraud at a Colombian private university. This information could help to conceive strategies to tackle academic fraud more broadly. To the best of our knowledge, this will be the first study using FCM to explore factors that promote or prevent academic fraud in education.

FCM will allow a participatory approach to identify factors related to academic fraud and to discuss potential solutions with stakeholders. Participatory research includes the promotion of action and contributes to address gaps in both knowledge and practice [25]. It involves co-design and co-ownership of health solutions between researchers and end-users. Collaboration with stakeholders ensures interventions that are aligned to the needs and specificities of end-users, thus improving their reach, adoption, and effectiveness [26]. We will discuss the study results with participants to identify priority areas to be addressed and to generate recommendations for action. Co-designed solutions will be implemented and further investigated through research designs such as a pilot randomised controlled trixsal (not described in this article).

FCM uses graphical language, which our previous research confirms is very accessible across educational and cultural backgrounds [27]. This will facilitate communication of the results across the university or with other groups. Obvious power disparities between faculty and students hinder discussion across the academic community. The use of cognitive maps representing the anonymous views of different stakeholder groups could facilitate dialogue among them. Because the maps do not contain any personal information, we expect that students will feel safe presenting their perspectives and proposals. Even more, we expect that reflecting on fraud could detonate internal dynamics that could promote students to reduce dishonest behaviors.

This project is a student-driven initiative. With the support of advisors (JP, CC, IS) the medical students co-authoring this manuscript (PL, JR, LN, DJ) defined the research question and research methods, and will guide data collection, analysis, and results dissemination.

### Limitations

The maps of students, interns, and resident might suffer social desirability bias [28] due to power imbalances that are present in medical education. To mitigate this, we will encourage trainees to be sincere when making the maps. We will let them know that they should not give responses they think we want to hear, but those that best reflect what they know. It is possible that students who engage in academic fraud will be less willing to participate as volunteers. Therefore, we need to interpret the maps as general ideas of what stakeholders think are the causes and not as cross-sectional evaluations of factors associated with fraud. Finally, we will inform trainees that their answers will not have any impact on their academic record. We are aware these measures do not preclude social desirability bias and we will interpret our results with caution.

Despite our plan for collecting data until we reach saturation of map concepts, our results might represent only a portion of the potential opinions across the faculty and are not generalizable to students in other universities. The maps are intended to facilitate deliberative dialogue among the academic community more than to provide a definite answer about fraud causes. During deliberative dialogues, stakeholders will provide deeper reflections on how the identified mechanisms could inform actions in the context of La Sabana University.

The definition of fraud that we will use during the mapping sessions does not differentiate between the seriousness of offenses, which could be linked to varying causes. We will discuss this with stakeholders to gain more insight into these discrepancies. We are aware that

academic fraud is context-specific, and our results might not be fully generalizable [6, 7]. However, the research methods we employ could be transferable to other settings in Latin America and elsewhere to facilitate deliberative dialogue among stakeholders.

## Conclusion

This study will shed light on the factors associated with academic fraud at a Colombian private university. This information will be an important input to conceive strategies to tackle academic fraud in the country. Our participatory research approach will likely boost the reach, adoption, and effectiveness of the intervention. We will also suggest factors that can be addressed in other contexts with similar problems.

## Supporting information

**S1 File. Advertisement to be used for the recruitment of participants.**
(DOCX)

## Author Contributions

**Conceptualization:** Juan Pimentel, Paola López, Johan Rincón, Laura Neira, Daniel Jiménez, Iván Sarmiento.

**Funding acquisition:** Juan Pimentel, Paola López, Johan Rincón, Laura Neira, Daniel Jiménez.

**Investigation:** Juan Pimentel, Paola López, Johan Rincón, Laura Neira, Daniel Jiménez, Camilo Correal, Iván Sarmiento.

**Methodology:** Juan Pimentel, Iván Sarmiento.

**Project administration:** Juan Pimentel.

**Software:** Iván Sarmiento.

**Supervision:** Juan Pimentel, Iván Sarmiento.

**Writing – original draft:** Juan Pimentel, Iván Sarmiento.

**Writing – review & editing:** Juan Pimentel, Paola López, Johan Rincón, Laura Neira, Daniel Jiménez, Camilo Correal, Iván Sarmiento.

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
