## [Decision Letter · Decision Letter 0]

24 May 2023

PONE-D-22-27129What facilitates or prevents academic fraud in a Colombian faculty of medicine: protocol of a study using fuzzy cognitive mappingPLOS ONE

Dear Dr. Pimentel,

Thank you for submitting your manuscript to PLOS ONE. After careful consideration, we feel that it has merit but does not fully meet PLOS ONE’s publication criteria as it currently stands. Therefore, we invite you to submit a revised version of the manuscript that addresses the points raised during the review process.

We look forward to receiving your revised manuscript.

Kind regards,

Sonia Vasconcelos, PhD

Academic Editor

PLOS ONE

Reviewers' comments:

Reviewer's Responses to Questions

**Comments to the Author**

1. Does the manuscript provide a valid rationale for the proposed study, with clearly identified and justified research questions?

Reviewer #1: Yes

Reviewer #2: Yes

2. Is the protocol technically sound and planned in a manner that will lead to a meaningful outcome and allow testing the stated hypotheses?

Reviewer #1: Yes

Reviewer #2: Yes

3. Is the methodology feasible and described in sufficient detail to allow the work to be replicable?

Reviewer #1: Yes

Reviewer #2: Yes

4. Have the authors described where all data underlying the findings will be made available when the study is complete?

Reviewer #1: Yes

Reviewer #2: No

5. Is the manuscript presented in an intelligible fashion and written in standard English?

Reviewer #1: Yes

Reviewer #2: Yes

6. Review Comments to the Author

You may also provide optional suggestions and comments to authors that they might find helpful in planning their study.

Reviewer #1: This is an interesting study addressing academic integrity. The proposed qualitative approach is very appropriate to explore the phenomenon of academic fraud in a university setting.

The protocol is well described and well worked out. I have a few comments to improve the presentation of the protocol:

1. It is not clearly stated whether the groups will be homogenous (a group with students, a separate one with professors, etc). This is important because the group dynamics would be different if the groups were heterogenous. Students may not be willing to talk openly about academic misconduct in front of their professors.

2. It is stated that the principal investigator will train four facilitators for the group exercise, but details are not provided about what this training will include.

3. The generalizability of the findings should be addressed.

4. Ethics and integrity considerations: as the group discussion may reveal sensitive information, or discover cases of academic misconduct. How will this be handled?

Reviewer #2: This is an interesting protocol with potential to generate relevant and useful results. The manuscript is well presented and I have few comments on it

1) The recruitment procedures are not clear. How are participants going to be "invited"? What will be the advertisement process? How do the authors plan on achieving a university-wide representative sample?

2) Since the study is based on volunteers, there is a risk that a preponderance of those more interested/knowledgeable on the subject will volunteer. Do the authors think this would imply on the risk of some bias? How do the authors plan to deal with it?

3) The authors use a broad definition of "academic fraud". Are the factors that prevent/promote fraud the same for minor vs major infractions? This appears to be a critical, implicit, assumption of the study.

4) I wondered why can't participants give their information in a more private fashion. For instance, they could drop their assessments inside a box which would then afterwards be retrieved by the researchers. Wouldn' that be feasible and more confortable for the participants? Is there any reason not to use this method?

5) Please do not use "Giving informed consent" as an inclusion/exclusion criterion. "Ça va sans dire".

6) In the methods: "The weights of the relationships in the group map will be the average of their weights in the

individual maps". When one works with averages it is usual to also have measures of dispersion. Does the analysis include something like that?

Again, I consider this a good study protocol with a good chance of success.

7. PLOS authors have the option to publish the peer review history of their article (what does this mean?). If published, this will include your full peer review and any attached files.

Reviewer #1: **Yes: **Ana Marusic

Reviewer #2: **Yes: **RENAN MORITZ VARNIER RODRIGUES DE ALMEIDA

---

## [Author Response · Author response to Decision Letter 0]

9 Aug 2023

July 20, 2023

Manuscript title: What facilitates or prevents academic fraud in a Colombian faculty of medicine: protocol of a study using fuzzy cognitive mapping

Dear Dr. Sonia Vasconcelos

PLOS ONE

With this letter, I am submitting the revised protocol, which includes the changes in response to the reviewers' comments. We appreciate the reviewers' feedback and are confident their suggestions have enhanced our paper. 

The manuscript complies with PLOS ONE style requirements. We also reviewed the list of references to ensure it is complete and correct. The ethics statement is now in the methods section. 

A point-by-point response to the reviewers' comments follows.

Kind regards,

Juan Pimentel PhD

Corresponding author

Reviewer #1:

1. It is not clearly stated whether the groups will be homogenous (a group with students, a separate one with professors, etc). This is important because the group dynamics would be different if the groups were heterogenous. Students may not be willing to talk openly about academic misconduct in front of their professors.

Andersson and Silver have also brought up this important concern during their explanation of FCM sessions. Pages 6, lines 122 to 125, explain that stakeholders will be categorized into homogeneous groups to participate in the mapping sessions.

2. It is stated that the principal investigator will train four facilitators for the group exercise, but details are not provided about what this training will include.

The revised text provides an explanation of the training process and includes references to the tools that will be utilized for training (page 7, lines 135 to 140).

3. The generalizability of the findings should be addressed.

In the limitations section, we clarified that the maps are not designed to provide a definite answer regarding the causes of fraud. Instead, they will be utilized to discuss additional ideas with the stakeholders at La Sabana University (page 14, lines 317 to 322). Furthermore, we explained that the maps are specific to the context, and therefore, we do not anticipate generalizing the results. Rather, we aim to offer an approach to encourage deliberative dialogue among stakeholders (page 14, lines 325 to 328).

4. Ethics and integrity considerations: as the group discussion may reveal sensitive information, or discover cases of academic misconduct. How will this be handled?

This is an important ethical concern that we now address on page 10, lines 234 to 237. During mapping sessions, participants are strongly encouraged to refrain from sharing personal experiences or any information that could potentially identify individuals. We won't delve into individual cases, and facilitators will encourage theoretical conversation about these topics. Also, we do not attempt to record the sessions or write any information that could lead to identify individuals. 

Reviewer #2: 

1) The recruitment procedures are not clear. How are participants going to be "invited"? What will be the advertisement process? How do the authors plan on achieving a university-wide representative sample?

Thank you for giving us the opportunity to clarify an important omission that we made. On page 6, lines 130 to 134, we included details on the recruitment process. You can locate a copy of the advertisement in Supplemental file 1.

2) Since the study is based on volunteers, there is a risk that a preponderance of those more interested/knowledgeable on the subject will volunteer. Do the authors think this would imply on the risk of some bias? How do the authors plan to deal with it?

Yes, this is an important consideration. Individuals who engage in fraudulent activities may be reluctant to participate in the sessions. However, it is important to note that the purpose of the maps is to reflect the opinions of stakeholders rather than personal experiences of each individual. Therefore, we anticipate that the maps will provide sufficient information to facilitate discussions at the faculty of medicine and guide future research. The limitations section now has a note to explain this (page 14, lines 312 to 315).

3) The authors use a broad definition of "academic fraud". Are the factors that prevent/promote fraud the same for minor vs major infractions? This appears to be a critical, implicit, assumption of the study.

We made explicit this limitation both in the methods and discussion sections (page 6, lines 113 to 116 and page 14 line 324).

4) I wondered why can't participants give their information in a more private fashion. For instance, they could drop their assessments inside a box which would then afterwards be retrieved by the researchers. Wouldn' that be feasible and more confortable for the participants? Is there any reason not to use this method?

Thank you for suggesting this reflection. The group mapping sessions aim to encourage dialogue among similar groups of stakeholders at an introductory level. This approach helps to keep the discussion at a theoretical level rather than focusing on specific cases or individual situations. These sessions are a part of a wider process within the faculty to promote reflection on academic fraud among the academic community. The group sessions serve as an initial stage of this process. We acknowledge that there may be a tendency for students to respond in a biased manner and have noted this as a potential limitation. However, if the goal is to highlight individual experiences or characteristics of fraud cases, other methods, including those mentioned in the comment, may be necessary. 

5) Please do not use "Giving informed consent" as an inclusion/exclusion criterion. "Ça va sans dire".

Thank you. We have reorganised the description of the inclusion criteria to make it more precise (page 6, line 29).

6) In the methods: "The weights of the relationships in the group map will be the average of their weights in the individual maps". When one works with averages it is usual to also have measures of dispersion. Does the analysis include something like that?

This is a valid point, and we appreciate this indication. We included a note explaining that we will provide a measure of dispersion to indicate the level of average within stakeholder groups (page 9 line 9).

---

## [Decision Letter · Decision Letter 1]

6 Sep 2023

What facilitates or prevents academic fraud in a Colombian faculty of medicine: protocol of a study using fuzzy cognitive mapping

PONE-D-22-27129R1

Dear Dr. Pimentel,

We’re pleased to inform you that your manuscript has been judged scientifically suitable for publication and will be formally accepted for publication once it meets all outstanding technical requirements.

Kind regards,

Sonia Vasconcelos, PhD

Academic Editor

PLOS ONE

Reviewers' comments:

Reviewer's Responses to Questions

**Comments to the Author**

1. Does the manuscript provide a valid rationale for the proposed study, with clearly identified and justified research questions?

Reviewer #1: Yes

Reviewer #2: Yes

2. Is the protocol technically sound and planned in a manner that will lead to a meaningful outcome and allow testing the stated hypotheses?

Reviewer #1: Yes

Reviewer #2: Yes

3. Is the methodology feasible and described in sufficient detail to allow the work to be replicable?

Reviewer #1: Yes

Reviewer #2: Yes

4. Have the authors described where all data underlying the findings will be made available when the study is complete?

Reviewer #1: Yes

Reviewer #2: Yes

5. Is the manuscript presented in an intelligible fashion and written in standard English?

Reviewer #1: Yes

Reviewer #2: Yes

6. Review Comments to the Author

You may also provide optional suggestions and comments to authors that they might find helpful in planning their study.

Reviewer #1: The authors addressed all my concerns and improved the manuscript accordingly. I do not have further comments.

Reviewer #2: The protocol seems now to be suitable for implementation; good luck with your study; glad to have helped

7. PLOS authors have the option to publish the peer review history of their article (what does this mean?). If published, this will include your full peer review and any attached files.

Reviewer #1: **Yes: **Ana Marušić

Reviewer #2: **Yes: **RENAN MORITZ VARNIER RODRIGUES DE ALMEIDA

---

## [Editor Report · Acceptance letter]

12 Sep 2023

PONE-D-22-27129R1 

What facilitates or prevents academic fraud in a Colombian faculty of medicine – protocol of a study using fuzzy cognitive mapping 

Dear Dr. Pimentel:

I'm pleased to inform you that your manuscript has been deemed suitable for publication in PLOS ONE. Congratulations! Your manuscript is now with our production department. 

Kind regards, 

on behalf of

Dr. Sonia Vasconcelos 

Academic Editor

PLOS ONE